

# Characteristics and source apportionment of

# fine haze aerosol in Beijing during the winter of 2013

Xiaona Shang[1], Meehye Lee[1], Fan Meng[2], Shihao Wang[2], Inseon Suh[1], Daegon Kim[3], Kwonho Jeon[3], Xuezhong Wang[2], Yuxi Zhao[2], Kai Zhang[2*]

[1] Department of Earth & Environmental Sciences, Korea University, Seoul, South Korea

[2] State Key Laboratory of Environmental Criteria and Risk Assessment, Chinese Research Academy of Environmental Sciences, Beijing 100012, China

[3] Department of Climate & Air Quality Research, National Institution of Environmental Research, Incheon, South Korea

Correspondence to: K. Zhang (zhangkai@craes.org.cn)



**Abstract**
For PM$_{2.5}$ filter samples collected daily at the Chinese Research Academy of Environmental
Sciences (Beijing, China) from December of 2013 to February of 2014 (the winter period),
chemical characteristics and sources were investigated with an emphasis on haze events in
different alert levels. During the three months, the average PM$_{2.5}$ concentration was 89 μg m$^{-3}$,
exceeding the Chinese national standard of 75 μg m$^{-3}$ in 24 h. The maximum PM$_{2.5}$
concentration was 307 μg m$^{-3}$, which characterizes *developed-type* pollution (PM$_{2.5}$/PM$_{10}$ >
0.5) in the World Health Organization criteria. PM$_{2.5}$ was dominated by SO$_4^{2-}$, NO$_3^-$, and
pseudo-carbonaceous compounds with obvious differences in concentrations and proportions
between non-haze and haze episodes. The non-negative matrix factorization (NMF) analysis
provided reasonable PM$_{2.5}$ source profiles, by which five sources were identified: soil dust,
traffic emission, biomass combustion, industrial emission, and coal combustion accounting
for 13 %, 22 %, 12 %, 28 %, and 25 %, respectively. The dust impact increased with
northwesterlies during non-haze periods and decreased under stagnant condition during haze
periods. A blue alert of heavy air pollution was characterized by the greatest contribution
from industrial emissions (61 %). During the Chinese Lantern Festival, an orange-alert was
issued and biomass combustion was found to be the major source owing to firecraker
explosions. Red-alert haze was almost equally contributed by local traffic and transported
coal combustion emissions from Beijing vicinities (approximately 40 % each) that was
distinguished by the highest levels of NO$_3^-$ and SO$_4^{2-}$, respectively. This study also reveals
that the severity and source of haze are largely dependent on meteorological conditions.

Key words: PM$_{2.5}$, winter haze, Beijing, chemical composition, source apportionment, NMF





## 1. Introduction

With the increasing $PM_{2.5}$ concentration in northern China, winter haze occurrences increased from 3 to 16 days during 2000–2012 (Wang and Chen, 2016). The frequency of haze events during winter is enhanced by meteorological conditions; the minimum daily temperatures typically reach −15 to −20 °C (Wu et al., 2012) and the boundary layer height becomes shallow (Zheng et al., 2015). Moreover, the combustion of fossil fuel increases at low temperatures (Zhang and Samet, 2015). As the air quality deteriorated, China released its third revision of the "The National Ambient Air Quality Standards" (NAAQS) in 2012 (GB 3095-2012), which stipulated safe $PM_{2.5}$ levels for the first time (Zhang and Cao, 2015). However, the worst haze events in the major cities of China were recorded during the winter of 2012–2013. During January of this period, Beijing experienced almost daily haze and the hourly $PM_{2.5}$ concentration reached 855 µg m$^{-3}$ (Zheng et al., 2015). In Beijing, winter haze usually lasts for approximately five days (Zheng et al., 2015, 2016). The long duration of haze with high $PM_{2.5}$ concentration triggers a red alert for air pollution (Liu et al., 2017), which is the highest level of the heavy air polllution warning system issued in the "Emergency plan for heavy air pollution in Beijing (revised in 2016)" (in Chinese: http://zhengce. beijing.gov.cn/library/192/33/50/200/806828/96701/index.html).

The concentrations of $SO_2$, $NO_x$, and volatile organic compounds (VOCs), which are important precursors of $PM_{2.5}$, vary in different emission and policy implementations. Related particulate compositions (sulfate, nitrate, and organic matter) comprise two thirds of $PM_{2.5}$ (Huang et al., 2014; Hu et al., 2015). Over the past seven years (2000–2006), $SO_2$ emission has increased by 53 %, consistent with the increases in power plant emissions from 10.6 Tg to 18.6 Tg (Lu et al., 2010). Particularly in northern China, the emissions from power plants have increased by 85 %. In contrast, $SO_2$ levels have significantly decreased since 2006, when stricter $SO_2$ regulations, such as the use of flue-gas desulfurization systems or scrubbers, were imposed (Van der A et al., 2016). The reduction was particularly rapid during 2008–2009. On the other hand, the $NO_x$ concentration increased from 2000 to 2012 (Hong et al., 2016; Cao et al., 2011). This increase is in accord with the increased number of vehicles, which contribute 90 % of the total $NO_x$ emissions in Beijing (Hendrick et al., 2014; Wu et al., 2012).



81 Meanwhile, the continuous increase in VOC emissions (from 13 Tg/yr in 2000 to 26 Tg/yr in

82 2012) was mainly driven by industrial processes (~70 %) (Hong et al., 2016). Coal

83 combustion (especially that of raw coal) from households is underestimated in the southern

84 and eastern rural areas of Beijing. Rural coal combustion comprises approximately 75 % of

85 Beijing's total coal combustion (Cheng et al., 2017). After the 2008 Olympic Games,

86 residential coal combustion emitted large amounts of $SO_2$, $NO_x$, and VOCs (70, 17, and 43 kt,

87 respectively). In 2013, these amounts had increased twofold to 132, 33, and 81 kt,

88 respectively. At the end of 2013, China issued the Air Pollution Prevention and Control

89 Action Plan (CAAC, 2013), which greatly reduced the precursor emissions in 2014 (Wang et

90 al., 2015).

91

92 Under the strict regulations on boiler and industrial emissions, $SO_2$ concentrations in Beijing

93 significantly decreased during the winter of 2013 and the fuel sulfur was reduced by more

94 than 80 % in 2014 (relative to its 2013 levels) (CAAC, 2013; CAAC, 2015). Over the same

95 period, the $NO_x$ levels were reduced by 6.7 % over the nation, but exceeded the standard by

96 42 % in Beijing, where local traffic emissions remained high. Meanwhile, the $PM_{2.5}$ pollution

97 is the most severe in the region of southern Beijing, where the annual average concentration

98 reached 150 μg m$^{-3}$ during 2014–2015. The level is comparable to the national standard of

99 $PM_{10}$ (CAAC, 2015; Zhang and Cao, 2015).

100

101 Since the 2008 Olympics and 2013 CAACs, heavy industries have been relocated and high-

102 quality fuel has been introduced. Both actions have reduced the concentrations of gaseous

103 precursors (Wang et al., 2009; Van der A et al., 2016), although these reductions are in

104 contrast to the frequent hazes currently observed in Beijing. In recent studies, the $PM_{2.5}$, dust,

105 and $SO_2$ concentrations in Beijing have been mainly attributed to regional transport (Wang et

106 al., 2014; Yang et al., 2013; Wang et al., 2011). Considering the extreme haze situation in

107 Beijing, researchers have sought the crucial factors of haze formation, usually by identifying

108 the emission sources of $PM_{2.5}$. The source apportionment of $PM_{2.5}$ is commonly analyzed by

109 source receptor models such as positive matrix factorization (PMF) and non-negative matrix

110 factorization (NMF) (Reff et al., 2007; Kfoury et al., 2016). These models have implicated

111 coal and industries as major sources of $PM_{2.5}$ in Beijing (Huang et al., 2014; Zhang and Cao,



2015; Zhang et al., 2013).

Following the severe and frequent haze occurrences in January of 2013, the chemical
characteristics and sources of $PM_{2.5}$ in Beijing were extensively investigated (Jiang et al.,
2015; Zheng et al., 2015; Zhang et al., 2015; Chen et al., 2017). However, few studies have
investigated the winter season of 2013–2014, which immediately followed the enactment of
the 2013 CAAC in China. In particular, the source apportionment of Beijing's haze remains
unknown (Wu et al., 2016). In the present study, we thoroughly examine the chemical
compositions of $PM_{2.5}$ in Beijing during the winter of 2013–2014, and accordingly, diagnose
the haze occurrence, probe the local and transported influence on haze, and quantify the
critical source contributions.

**2. Experiments**

Filtered samples of $PM_{10}$ and $PM_{2.5}$ were collected on the roof of a three-story container (~15
m above ground level) at the Chinese Research Academy of Environmental Sciences (CRAES)
in Beijing, China (40.04 °N, 116.42 °E), from December of 2013 to February of 2014. The
site is located near the four-way intersection of a residential area located between the 5[th] and
6[th] ring roads of Beijing.

Aerosols were collected for 24 hours (from 7 pm to 7 pm next day) on a 90-mm
polypropylene filter using a medium volume sampler at a flow rate of ~100 L/min (2030,
Laoying, China). Seventy $PM_{2.5}$ samples were collected and analyzed. The water-soluble ions
($Cl^-$, $NO_2^-$, $CO_3^{2-}$, $SO_4^{2-}$, $NO_3^-$, $Na^+$, $NH_4^+$, $K^+$, $Mg^{2+}$, and $Ca^{2+}$) were measured by ion
chromatography (IC25, Dionex, USA) with a detection limit between 0.01 and 0.06 μg m$^{-3}$.
The ionic measurement method is detailed in Lim (2009). For trace elemental analysis, the
samples were digested by a mixture of acids as described in Zhang et al. (2014). A quarter of
each filter was placed into a polytetrafluoroethylene flask and digested with 8 mL of
$HNO_3/H_2O_2$ (6/2 v/v, superpure grade, Merck, Darmstadt) at 180 °C for 8 h. The solution was
separated by centrifugation and diluted to 25 mL with ultrapure water. The concentrations of



trace metals (21 species, including Si) were determined by inductively coupled plasma-optical
emission spectrometry (Prodigy 7, Teledyne Leeman, USA). The mass concentration of $PM_{10}$
was also determined for comparison with that of $PM_{2.5}$.

The total concentrations of the water-soluble ions and trace elements were subtracted from the
$PM_{2.5}$ mass, to provide a measure that likely represents the carbonaceous components that
were not directly measured. In this study, therefore, it was referred as the pseudo-
carbonaceous components and used for the following discussion. The concentrations of these
pseudo-carbonaceous components were comparable to those of $PM_{2.5}$ concentrations observed
in Beijing (Ji et al., 2016). A meteorological suite of relative humidity, temperature, and
visibility was collected by CRAES from a sharing network of the China Meteorological Data
Service Center (CMDC): http://data.cma.cn/en/?r=data/detail&dataCode=A.0012.0001. The
gaseous species $NO_x$, $SO_2$, CO, and $O_3$ were measured using commercial analyzers (42i, 43i,
48i, 49i, Thermo Fisher, USA) in CRAES.

The $PM_{2.5}$ source was identified by non-negatice matrix factorization (NMF) analysis.
Introduced by Lee and Seung (1999, 2001), NMF operates similarly to positive matrix
factorization (PMF). Both analysis methods find two matrices (W and H, termed the
contribution matrix and the source profile matrix, respectively) that best reproduce the input
data matrix (V) using the same factorization approach (V = WH) as a positive constraint.
However, while PMF is a generalized, alternative least-squares method, NMF minimizes the
conventional least-squares error and the generalized Kullback–Leibler divergence. The
uncertainties in NMF analysis were estimated as 0.3 + the analytical detection limit (Xie et al.,

165 1999).


In addition to NMF analysis, the origin of air masses were traced by trajectory analysis. For
air masses arriving at 500 m altitude, backward trajectories were computed for 72 hours using
HYSPLIT model with GDAS data in SplitR (Stein et al., 2015, https://github.com/rich-
iannone/ SplitR).





## 3. Characteristics of winter PM$_{2.5}$

### 3.1. PM$_{2.5}$ and PM$_{10}$ mass variations

During the 2013–2014 winter period in Beijing, the mass concentrations of PM$_{2.5}$ and PM$_{10}$ varied in a similar pattern (Fig. 1). Zheng et al. (2015) reported a similar trend between the PM$_{2.5}$ and PM$_{10}$ concentrations. In this study, the average PM$_{10}$ concentration was 142 μg m$^{-3}$, comparable to the Chinese national standard of 150 μg m$^{-3}$ in 24 h (secondary standard of GB 3095-2012). However, the mean PM$_{2.5}$ concentration was 89 μg m$^{-3}$, exceeding the standard of 75 μg m$^{-3}$ in 24 h. The PM$_{2.5}$ standard was most severely exceeded in February 2014, when the average concentration (133.5 μg m$^{-3}$) reached the highest winter concentration in Beijing during the 2005–2015 decade (Lang et al., 2017).

Based on the criteria of the World Health Organization (WHO) (2006), the wintertime air pollution of Beijing was classified as *developed-type*, meaning that the PM$_{2.5}$/PM$_{10}$ ratio exceeded 0.5 in 70 % of the samples (Table 1). The mean PM$_{2.5}$ concentration of these samples (113 μg m$^{-3}$) was four times higher than that in *developing-type* pollution (31 μg m$^{-3}$). In approximately half of the *developed-type* samples, the PM$_{2.5}$ and PM$_{10}$ mass concentrations exceeded the national standards, all of which were collected during haze events. The average PM$_{2.5}$ concentration over 13 haze days reached 198 μg m$^{-3}$ and the visibility was significantly reduced to ~1 km (Fig. 1). In contrast, the PM$_{2.5}$ concentration exceeded the standard without violating the PM$_{10}$ concentration on only a few days. These results well reflect the wintertime characteristics of PM$_{2.5}$ levels in Beijing, which are largely related to haze episodes. The average PM$_{2.5}$ concentration of the *developed-type* was comparable to that of the *developing-type* unless the PM$_{2.5}$ concentration exceeded the standard.

On 12 out of 13 haze days, the pollutant levels met the criteria of heavy air pollution alerts stipulated in the "Emergency plan for heavy air pollution in Beijing (revised in 2016)". In the lowest level of the four-tier warning system, blue alert, the daily average air quality index (AQI) exceeded 200 on only one day. In Table 1, the one no-alert and three blue-alert haze days are defined as no/blue-alert haze events. The average PM$_{2.5}$ concentration on these days was 168.4 μg m$^{-3}$ (Table 1). During the red-alert period (February 20–25), the daily PM$_{2.5}$





concentration peaked at 305.6 μg m$^{-3}$. A red alert is declared when the air pollution is heavy
and severe. During a red alert, AQI exceeds 200 on four consecutive days and exceeds 300 on
continuous two of those days. Although the daily average AQI remained higher than 300
during the February 14–16 period, this event was an orange alert because it continued for only
three days. The AQI data can be found at http://www.tianqihoubao.com/aqi/beijing-
201402.html (in Chinese). Here, we describe episodes in terms of alerts defined in the heavy
air pollution system rather than in the haze alert system, because the former definition is
based on the daily averaged AQI, whereas the three-tier haze warnings depend on the hourly
meteorological parameters (relative humidity and visibility) or PM$_{2.5}$. Because we measured
the daily concentrations, the heavy air pollution alert was suitable for our purpose.
**3.2. Chemical composition**

Throughout the wintertime, the average PM$_{2.5}$ concentration remained close to 90.0 μg m$^{-3}$,
20 % above the national standard. The major PM$_{2.5}$ components were SO$_4^{2-}$, NO$_3^{-}$, NH$_4^{+}$, and
pseudo-carbonaceous compounds, with average concentrations of 18.8, 16.9, 8.5, and 38.6 μg
m$^{-3}$, respectively. Collectively, these four compositions comprised 83 % of the PM$_{2.5}$ mass
(Fig. 2). On the 57 non-haze days, the fractional chemical compositions and concentrations of
SO$_2$ and NO$_2$ were comparable to those of the entire period (70 days). In contrast, the portions
of soil minerals such as Ca$^{2+}$ and trace elements (including Si) were 3–4 times higher on non-
haze days than on haze days. The Ca$^{2+}$ and Si concentrations were highly correlated ($r^2 = 0.8$)
and were more related to the PM$_{10}$ ($r^2 = 0.6$) than PM$_{2.5}$ levels. This reflects the significant
impact of soil dust on non-haze days (Fu et al., 2012). On haze days, the particle masses,
compositions, SO$_2$, and NO$_2$ varied widely among the different alert levels.

**3.3. Source profiles**

The PM$_{2.5}$ sources were identified in an NMF analysis of the measurement data. The data
included 8 water-soluble ions, 13 trace elements, and pseudo-carbonaceous compounds. After
comparison through a principle component analysis, the principal factors were determined.



Finally, five critical factors were distinguished: soil dust, traffic emission, biomass
combustion, industrial emission, and coal combustion (Table 2). The five source profiles are
presented in Figure 3. Despite their clear signatures, the contributions of dust and traffic
emissions were approximately half those of biomass combustion, industrial emission, and
coal combustion (Table 2).

Factor 1 (soil dust) is confirmed by high $Ca^{2+}$, Si, Fe, $Cl^-$, and $Na^+$ contents (Fu et al., 2012).
The high concentrations of $Cl^-$ and $Na^+$ likely originate from dry lake deposits (Abuduwaili et
al., 2015), which spread over the northern area of Beijing. Elevated heavy metals suggest the
presence of fugitive dust mixed with industry or traffic emissions (Wan et al., 2016). The high
loadings of $NO_3^-$ and $NH_4^+$ in Factor 2 indicate traffic emissions (He et al., 2016). As is well
known, $NH_3$ is emitted from three-way catalytic converters in vehicles (Chang et al., 2016).
Factor 3 (biomass combusion) emits large amounts of $K^+$ and $NH_4^+$ (Balasubramanian et al.,
1999), along with the elements that give exploding fireworks their color (namely Mg, Fe, Al,
Ti, Cu, and Si) (Baranyai et al., 2015). The concentrations of these firecracker indicators are
most significantly elevated during the Chinese Lantern Festival (14, 15, and 16 of February;
Fig. 1). Factor 4 (industrial emissions) is distinguished by high pseudo-carbonaceous
materials and heavy metals. Factor 5 (coal combustion) is characterized by high $Cl^-$, $SO_4^{2-}$,
and $NO_3^-$ contributions, which are absent in Factor 4. Although both Factors 4 and 5 represent
the influence of industrial emissions near Beijing, Factor 5 is more clearly sourced from
industries requiring high energy, such as iron and steel, cement, and power plants (Tan et al.,
2016; Zhang et al., 2013). In contrast, Factor 4 indicates emissions from industrial processes
using VOCs as raw materials (Yu et al., 2013; Wu et al., 2015).

In a previous study, source apportionment by NMF or PMF analysis distinguished 7–8 factors
(Zhang et al., 2013), including a secondary formation source. The secondary source was not
separated as an individual factor in the present study. As a typical secondary species, $SO_4^{2-}$
dominates in Factor 5. However, a $NO_3^-$ signature appears in all factors except Factor 4. This
study was performed in winter, during which the chemical composition of $PM_{2.5}$ was likely to
be more dependent on source strength rather than photochemical oxidation, generating
secondary species. Therefore, these five factors primarily indicates direct emission sources. In





addition, $NO_2$ is more likely sourced from local emissions, but $SO_2$ is expected to be
transported from nearby regions.

**4. Characteristics of winter haze**
**4.1. Chemical and meteorological characteristics**

The chemical compositions of $PM_{2.5}$ clearly differed on haze in contrast to non-haze days in
terms of secondary ions and pseudo-carbonaceous compounds (Fig. 2). The largest fraction of
pseudo-carbonaceous compounds (61 %) was accompanied with the smallest proportion of
$SO_4^{2-}$ (4%) on no/blue-alert days, suggesting low coal consumption by high-VOC-emitting
industries. On orange-alert haze events, the $NO_3^-$ fraction was twice that on non-haze days,
and the $K^+$ and $Mg^{2+}$ proportions were maximinized (at 6% and 1%, respectively), implying
biomass-combustion emission during the Lantern festical in China. The concentrations of
$SO_4^{2-}$ and $NO_3^-$ were comparable with the greatest contribution in red-alert haze events. In
addition, these species were closely related to the $Cl^-$ ($r^2 = 0.8$) and $NH_4^+$ ($r^2 = 0.9$)
concentrations, respectively, suggesting large contributions by coal combustion and vehicle
emission. It is also noteworthy that the $SO_4^{2-}$ fraction varied more widely than the $NO_3^-$
fraction. Among the three levels of haze events, $SO_4^{2-}$ varied from 4 % to 32 %, whereas $NO_3^-$
varied from 16 % to 31 % and $NH_4^+$ from 9 % to 11 %. Similarly, although both $SO_2$ and $NO_2$
concentrations were the highest in red-alert haze, $SO_2$ enhancement (relative to non-haze days)
was 20 % larger than $NO_2$ enhancement. Because the sulfur compounds were much more
elevated than the nitrogen compounds on haze days (particularly in red-alert haze events), the
winter haze in Beijing was concluded to be largely contributed by coal combusion, which
emits sulfur compounds. Furthermore, coal emissions are mostly transported from nearby
Beijing (Hendrick et al., 2014).

To examine the meteorological conditions favorable for haze occurrence and clarify the
emission source regions, surface weather maps combined with daily average backward
trajectories at 500 m were compared during non-haze and haze events. Previous studies also
reported that weather conditions were critical for haze formation. In East China, migratory
anticyclones and weak pressure gradients were the prerequisites of winter haze from 1980 to



2012 (Peng et al., 2016). High $PM_{2.5}$ episodes in Beijing usually began with weak southerly
winds and ended with strong northerly winds (Guo et al., 2014). In the present study, air mass
transported from the northwest shifted westward, and then to the southwest and southeast
regions under the migration of high pressures. Throughout this process, the weather condition
became increasingly stagnant (Fig. 4) and the haze-alert level increased gradually. When air
masses were rapidly transported from the northern desert area (Fig. 4a), mineral species such
as $Ca^{2+}$ and Si were enriched on non-haze days and the $PM_{10}$ mass was high. In the western
regions of Beijing (Fig. 4b), where various industries manufacture food, drink, furniture,
phamarceuticals, and other products from VOCs (http://www.berkeleysg.com/2016/06/china-
manufacturing-distribution-map/), the fraction of pseudo-carbonaceous compounds rose to its
maximum as the air mass slightly lingered over the region. During February 14–16,
firecracker explosions caused a spike in $K^+$, $Mg^{2+}$, and $NH^{4+}$ concentration under the stagnant
weather condition, in which the air mass moved very slowly from the southwestern areas,
where population density is the highest (Cheng et al., 2017). As the air mass moved eastward
toward the high energy-requiring regions (http://berc.berkeley.edu/energy-access-developing-
parts-china/) (Fig. 4d), such as Tianjin and Tangshan, where coal consumption is high (Cheng
et al., 2017), the $PM_{2.5}$ and $SO_4^{2-}$ ($SO_2$) concentrations reached their maxima.

**4.2. Source profiles**

To quantify each source contribution during the winter haze in Beijing, daily samples were
analyzed by NMF and the source profiles during haze and non-haze episodes were compared
(Fig. 5). In all samples, the main contributions were industrial, traffic, and coal combustion
emissions (22–28 %), followed by soil dust and biomass combustion (13 % and 12 %,
respectively). However, soil dust loading, which is associated with elevated fractions of Ca,
Si, and pseudo-carbonaceous matters (Fig. 2), was enhanced to 20 % during non-haze events.
Meanwhile, the local traffic contribution decreased as the air mass was rapidly transported
from the northwestern desert areas, as mentioned in subsection 4.1.

The three types of haze episodes exhibited strong contrasts not only in their chemical species
and source regions, as mentioned above, but also in their source profiles (Fig. 5). No/blue-



alert haze was dominated by industrial emissions (61 %) as the airflow passed over the
industrial regions manufacturing products from raw VOCs. Consequently, the pesudo-
carbonaceous concentration increased. During orange- and red-alert haze events, the dust
contribution was negligible and the anthropogenic fraction increased sharply. During the
Chinese Lantern Festival (which triggered an orange alert), a biomass signature with the
highest $K^+$ concentration was observed in the air mass transported from the southwestern
populated areas of Beijing. The $K^+$ contribution (35 %) was three times higher than that on
non-haze days. During Febuary 20–25, the outflow of the high coal-consuming eastern region
ehanced the proportion of coal combustion products to 37 %. Simultaneously, the traffic
contribution was the highest at 43 %. The coal and traffic effects were accompanied by two-
fold elevations of $SO_4^{2-}$ and $NO_3^-$ in $PM_{2.5}$.

**5. Policy implications**

During the 2013–2014 winter peiod in Beijing, the average $PM_{2.5}$ concentration exceeded the
standard by 20 %, and in February, reached its highest level in the 2005–2015 decade (Lang
et al., 2017). The $PM_{2.5}$ mass closure and concentration of gaseous precursors during the 57
non-haze days were comparable to those of the entire winter period. Mineral dust is an
important source of $PM_{2.5}$ and elevates the $PM_{10}$ concentration on non-haze days. The average
$PM_{2.5}$ concentrations increased significantly from 64.8 µg m$^{-3}$ on non-haze days to 168.4 µg
m$^{-3}$ on no/blue-alert days and to 217.7 µg m$^{-3}$ on red-alert days.

When weather conditions stagnate under weak pressure gradients, the alert levels of heavy air
pollution upgrade on haze days. The migratory anticyclones also shift the air masses, causing
wide variations in chemical species and emission sources. During haze days, the $NO_2$ and
$NO_3^-$ concentrations exceed those of $SO_2$ and $SO_4^{2-}$, respectively, but the sulfur-containing
species vary more widely than the nitrogen species. The sulfur compounds are particularly
enhanced in stagnant air masses transported from the Beijing vicinities, including the southern
and eastern regions, leading to the large sulfur variation with little change in nitrogen. These
results highlights the significant influence of the emissions from industries requiring high
energy and using coal in Beijing vicinities and from local vehicles on winter haze formation



in Beijing, which is in accordance with findings from previous studies (Hendrick et al., 2014;
Wang et al., 2016). To abate the severe haze in Beijing, therefore, it is necessary to reduce
vehicle emissions in Beijing and further sulfur emissions from industrial complexes in
surrounding cities. For cost-effectiveness, the weather forecast needs to be incoporated into
the policy implementation.

**6. Conclusion**

This study investigated the chemical characteristics of $PM_{2.5}$ during the 2013–2014 winter
period in Beijing and identified its sources with an emphasis on haze events by measuring the
particle masses, water-soluble ions, and trace elements in filtered samples. Finally, policy
implications for controlling haze occurrences in Beijing were deduced from the analysis.

The samples were collected daily at CRAES, Beijing, China, from December of 2013 to
February of 2014. During the winter period, the overall average $PM_{2.5}$ concentration in
Beijing was 89 μg/m$^3$, exceeding the Chinese national standard of 75 μg/m$^3$ in 24 h. The
excess was linked to high occurrence of haze events in February of 2014. The high $PM_{2.5}$
episodes were concurrent with $PM_{10}$ exceedence. Seventy percent of the samples were
identified as *developed-type* in the WHO criteria; that is, their $PM_{2.5}/PM_{10}$ ratios exceeded 0.5.
All 13 recognized haze events in this study were included in the *developed-type*.

The chemical compositions showed that secondary ions were doubled on haze days relative to
non-haze days, but mineral species were halved during haze events. For the 70 daily $PM_{2.5}$
samples, NMF analysis was performed and the source profiles were compared between haze
and non-haze days. The analysis identified five principle sources, of which industrial emission,
coal combustion, and traffic emission comprised similar fractions of 28 %, 25 %, and 22 %,
respectively. The soil-dust and biomass-combustion sources were well distinguished and
contributed 13 % and 12 %, respectively. Comparing the soure profiles between non-haze and
haze events, the impact of soil dust was most noticeable on non-haze days, when the air
masses rapidly transported from northwestern desert areas and brought high concentrations of


Ca$^{2+}$ and Si into Beijing. However, nearby transport of industrial, biomass combustion, and
coal combustion emissions, along with local traffic emission, contributed to haze events under
stagnant weather conditions. The contributions of these four sources increased by up to 61 %,
35 %, 37 %, and 43 % in no/blue-alert, orange-alert, and red-alert days, respectively. The
industries that are mainly located to the west of Beijing use VOCs as raw materials, elevating
the pseudo-carbonaceous components in PM$_{2.5}$. Biomass combustion increases during the
firework displays of the Lantern Festival (February 14–16). At that time, the K$^+$ and Mg$^{2+}$
concentrations are maximized. When a red-alert was issued for six days in 2014, the
contribution of SO$_4^{2-}$ and NO$_3^-$ increased by factors of 3 and 2, respectively, from their non-
haze levels. Overall, the sulfur compounds (SO$_2$ and SO$_4^{2-}$) varied much more widely than the
nitrogen compounds (NO$_2$ and NO$_3^-$) through haze events, implying the substantial
contribution of industrial emissions from coal combustion in surrounding cities. The high
level of nitrogen compounds suggests local vehicle emissions as a main source of winter haze
in Beijing. This study also emphasizes the role of weather condition in haze formation by
building up stagnant condition that facilitates the transport of industrial emissions from
Beijing vicinities. These findings will be applicable to policy making.


**Acknowledgements**


We thank the China·Korea Air Quality Joint Research Team for promoting two-side
cooperation. Special thanks to CRAES members, including Pengli Duan, Fenmei Xia,
Hongjiao Li, Zilong Zheng, Jing Zhou, Qingshu Ke, Jiaying Yang, and Jikang Wang, for
providing support for filter sampling. We also thank the funding support from Ministry of
Environment and National Institute of Environmental Research (Q1432252 and Q1432253),
South Korea; the National Natural Science Foundation of China (No. 41205093); the National
Department Public Benefit Research Foundation (No. 201109005); and the Fundamental
Research Funds for Central Public Welfare Scientific Research Institutes of China (No.
2016YSKY-025).



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



**Table 1.** Statistics of $PM_{2.5}$ mass concentrations.

| $PM_{2.5}$ mass classification | Number | $PM_{2.5}^{*}$ [µg m$^{-3}$] |
|---|---|---|
| Chemical and NMF analysis | 70 | |
| Comparison with $PM_{10}$ mass | 67 | 88.7 |
| $PM_{2.5}/PM_{10} > 0.5$ | 47 | 113.4 |
| $PM_{2.5} > 75$ µg m$^{-3}$ and $PM_{10} > 150$ µg m$^{-3}$ | 23 | 167.8 |
| $PM_{2.5} > 75$ µg m$^{-3}$ and $PM_{10} < 150$ µg m$^{-3}$ | 5 | 113.4 |
| $PM_{2.5} < 75$ µg m$^{-3}$ and $PM_{10} < 150$ µg m$^{-3}$ | 19 | 38.9 |
| $PM_{2.5}/PM_{10} \leq 0.5$ | 47 | 30.8 |
| Haze days[#] | 13 | 198.3 |
| Red alert | 6 | 217.7 |
| Orange alert | 3 | 216.2 |
| No/blue alert | 4 | 168.4 |

* Average concentration
[#] Heavy air pollution alert



**Table 2.** Sources identified by NMF analysis.

| Factor | Contribution | Sources |
|--------|-------------|---------|
| Factor 1 | 13 % | Soil dust |
| Factor 2 | 22 % | Traffic emission |
| Factor 3 | 12 % | Biomass combustion |
| Factor 4 | 28 % | Industrial emission |
| Factor 5 | 25 % | Coal combustion |





**Figure 1**. Variations in mass and chemical compostions of $PM_{2.5}$, $PM_{10}$ mass, gaseous precursors, and meteorological parameters measured from Dec. 1, 2013 to Feb. 28, 2014. Horizontal lines indicate the Chinese national standards of PM concentrations in 24 h and the vertically shaded regions denote the 13 haze days.




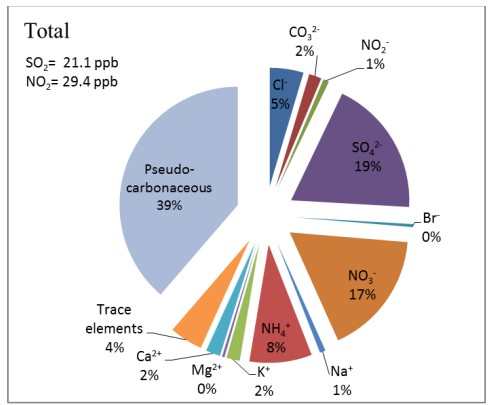

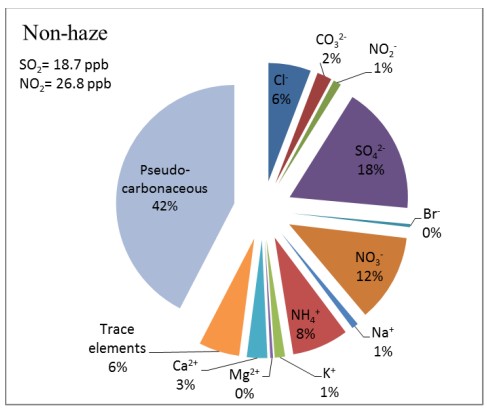

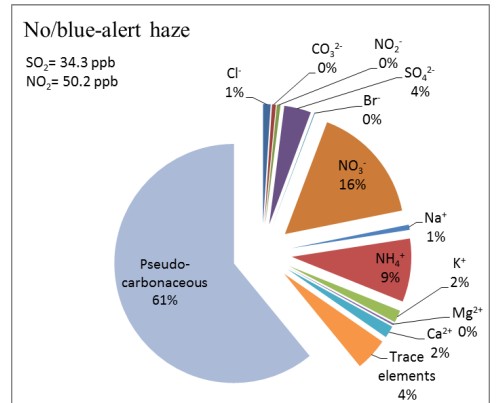

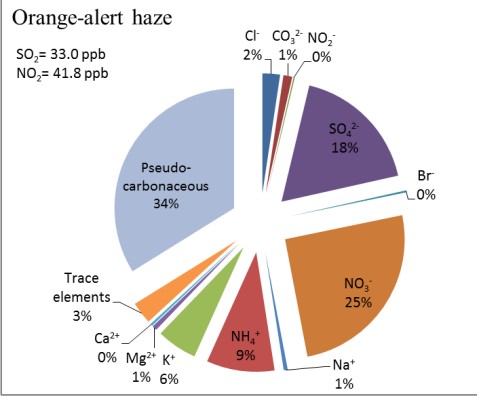

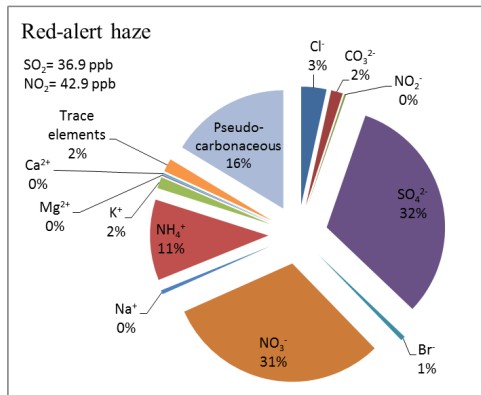

587

**Figure 2.** PM$_{2.5}$ mass contributions of water-soluble ions, trace elements, and pseudo-carbonaceous matter during the entire period (top left), non-haze days (top right), and haze days at blue-alert (center left), orange-alert (center right), and red-alert (bottom left) warning levels. The average SO$_2$ and NO$_2$ concentrations are also given.




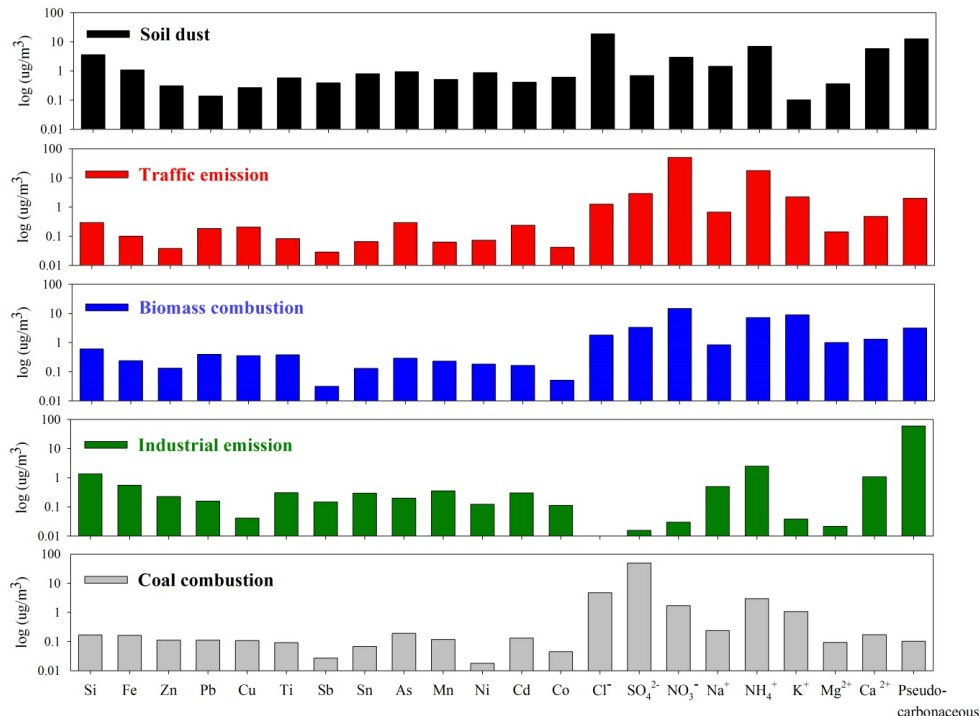

592

**Figure 3.** Composition profiles of the five factors identified in NMF analysis.

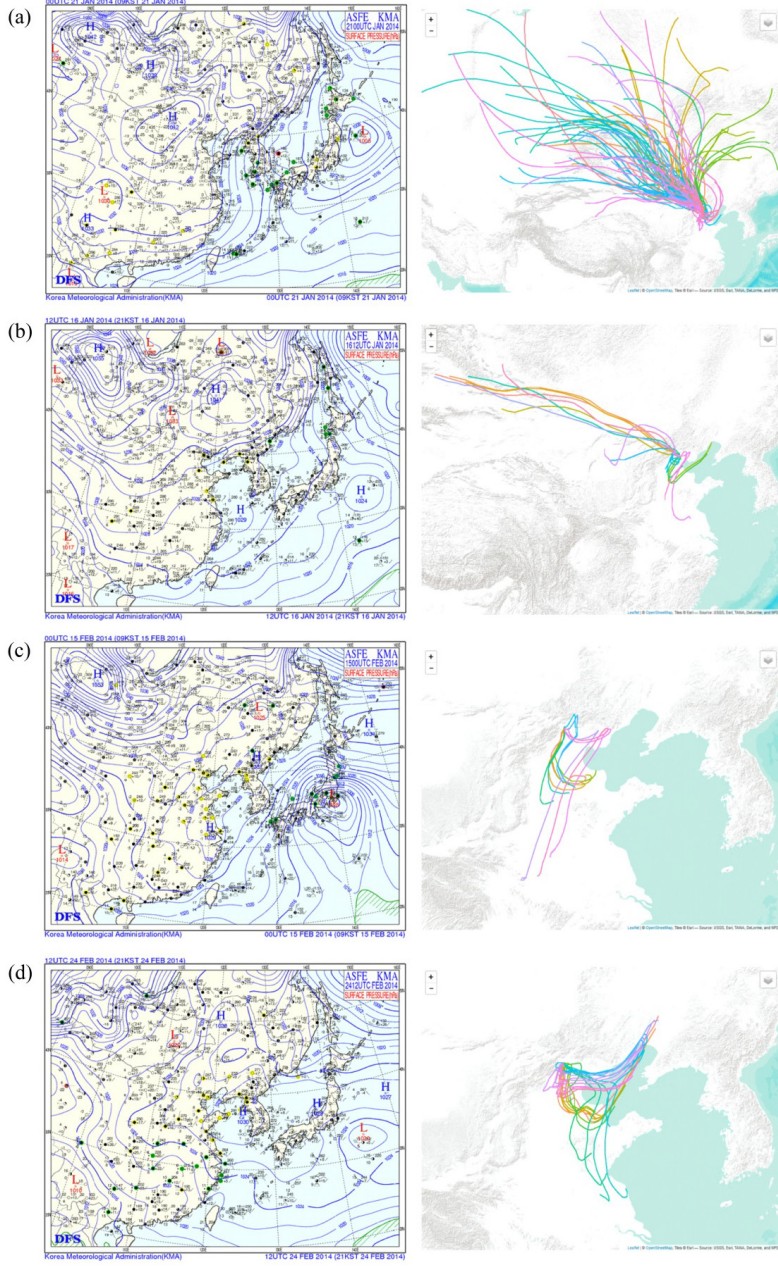

594

**Figure 4**. Surface weather maps and 72-h backward trajectories on days of (a) non-haze (57

days), (b) no/blue-alert haze (4 days), (c) orange-alert haze (3 days), and (d) red-alert haze (6

days). Trajectories were calculated twice a day at 18 and 06 UTC for non-haze days in (a)

and every 6 hours at 12, 18, 24, and 06 UTC for haze days in (b), (c), and (d).





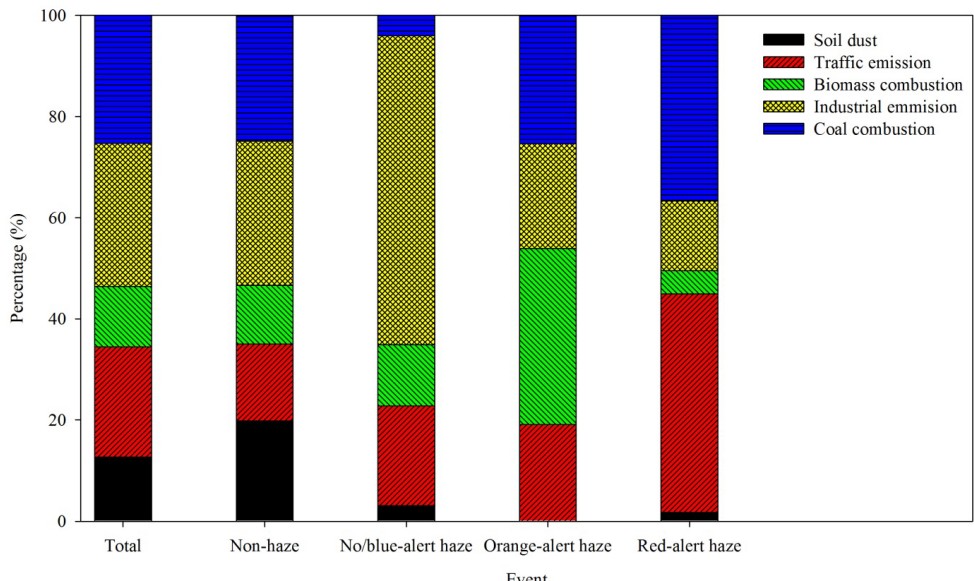

599

**Figure 5.** Comparison of source contributions (left to right) over the entire winter, during
non-haze events, and during no/blue-alert, orange-alert, and red-alert haze events.