# Peer review of "Characteristics and source apportionment of"

_Atmospheric Chemistry and Physics, 2017_

## Referee Comment (RC1) · Anonymous Referee #1 · 7 Jul 2017

General Comments:

In the current work, the authors presented a time series of filter measurements of PM chemical composition at a northern suburban site in Beijing. The measurement documented the serious winter pollution in Beijing. And the authors performed a NMF source apportion analysis of this dataset. They found Traffic emissions, Coal combustion, Industrial emission, Biomass combustion, and Soil dust were the five factors that contribute to the PM pollution. This work can serve as a useful document of that seriously polluted winter in Beijing 2013. I have the following comments for the authors to consider.

0Printer-friendly version

[Figure]

NMF is an approach which is less widely used by the community of PM source appointment than PMF or PMF/ME-2. It would be useful for the readers to judge the quality of the analyzed results if the authors could provide more details about the possible difference between NMF and PMF in the part of methods. It is well known that the use of such kind of statistical analysis tool is quite arbitrary. There are some plausible interpretations about the extracted factors in the paper. But please add uncertainty analysis of the NMF results.

The authors concluded that "To abate the severe haze in Beijing, therefore, it is necessary to reduce vehicle emissions in Beijing and further sulfur emissions from industrial complexes in surrounding cities." But this is not fully supported by the data presented in this work. Can you prove that local emissions are dominated by vehicles? Can you prove that sulfur emissions are mainly from industrial complexes in surrounding cities? How about the uncontrolled coal burning for sulfur emissions?

Specific Comments:

1. In the part of introduction, the authors may add descriptions on the current alert system implemented in Beijing. 2. The term "pseudo-carbonaceous" in Figure 2 and other place of corresponding text sounds strange. Maybe the authors can use "Particulate organic matter". 3. Line 260 – 261 "This study was performed in winter, during which the chemical composition of PM2.5 was likely to be more dependent on source strength rather than photochemical oxidation," this argument is ambiguous. The secondary species like NO3- and SO42- must come from atmospheric oxidation processes. I think even in winter chemical composition of PM2.5 was also related to both source strength and oxidations. Also as shown in Figure 2, sulfate and nitrate were always dominate chemical compositions especially for the conditions of pollution episodes. 4. Line 262 – 264 "In addition, NO2 is more likely sourced from local emissions, but SO2 is expected to be transported from nearby regions." This is a good argument. But more discussions or evidences are required to support this argument.
5. Line 273, what could be the high VOCs emitting industries? Please be more specific.

Technical Comments:

Line 202, 203 et al., I suggest the authors to present the concentrations of PM consistently for the significant figure as Line 177, 180 and 187, e.g. change 168.4 $\mu$g/m3 to 168 $\mu$g/m3.

---

## Referee Comment (RC2) · Anonymous Referee #2 · 17 Sep 2017

General Comments:

This paper present aerosol chemical composition, total aerosol mass and major gas phase air pollutants during the winter of 2013-2014 in Beijing. This winter followed the severe haze events that have been reported previously in the preceding year, in January of 2013, and also followed implementation of emissions reductions in between the two years. The authors use an NMF analysis to identify the major source factors that influence aerosol composition, and sort the data according to non-haze events and increasing severity of haze events that are defined by the Chinese national air quality standards.

[Figure]

The analysis provides information about total aerosol mass, its composition and its likely source contributions, both chemically and spatially, with respect to meteorological backward trajectories associated with each category. The information should be of interest to the readership of ACP and should be published following the authors attention to the comments below.

Specific comments:

Page 3, line 53: Does this mean 3 – 16 days per year?

Page 3, line 56: Is there a quantitative estimate for the boundary layer depth?

Page 3, line 62: suggest phrasing: "winter haze episodes are 5 days in duration"

Page 3, line 72: replace "Over the past seven years (2000-2006)" with "Over a seven-year period (2000-2006)". Then on line 75, add "... by 85% over this period".

Page 6, line 157: negative rather than negatice

Page 6, line 164: The uncertainty description is not clear. What are the units on "0.3 + the analytical detection limit" ? Is this a relative error, or does it have concentration units ?

Page 7, lines 178-179: What is meant by "secondary standard of GB 3095-2012" ?

Table 1: Should the number of days with PM2.5/PM10 > 0.5 and < 0.5 add up to the total number of days with comparison to PM10? In other words, 47 + 47 does not equal 67. The text implies that it should (e.g., that 70% of the events were developed type, which would be 47/67). Is the correct number for PM2.5/PM10 < 0.5 = 20 ?

Figure 3: The factors are shown on a log scale to illustrate the contributions from all of the components of chemical composition. However, the log scale hides the large contributions of individual components to each, such as sulfate to coal combustion. Can the figure also be shown on a linear scale for comparison to illustrate which components make large contributions to each factor? A linear scale would increase the

contrast.

Page 9, lines 241-243: Traffic is attributed to a factor with high nitrate an ammonium, with the ammonia precursor attributed to the same emission source as NOx, presumably. Should there also be an agricultural factor for the ammonia emissions? Can the authors comment?

Page 9-10, lines 256-264: The authors suggest that secondary production is a relatively unimportant consideration. However, it is well known that sulfur oxidation rates in winter are typically slow, while NOx oxidation rates to $NO_3^-$ can remain rapid (e.g., Calvert et al., Nature 1985). Can the authors comment on the source of sulfate? Does this likely arise from secondary oxidation of $SO_2$, or does it rather come from a primary emission of more oxidized sulfur that leads to sulfate? An easy metric here would be the ratio of sulfate to $SO_2$ in molar units. A similar comparison could be given for $NO_3^-$ to NOx.

Page 10, line 271: A large carbonaceous component is shown for blue / no alert days. However, there are only 4 days and 4 samples in this category. Is it possible that the deviation of the carbonaceous aerosol from the trend of decreasing contribution as the haze level increases is simply a result of the small number of samples in the blue / no alert category, leading to a statistically anomalous result? Can the authors comment on this?

Page 10, line 283 – 287: Following from the comment above, how does the sulfate / $SO_2$ ratio vary as the haze alert level increases? Does this ratio increase, decrease, or stay the same? If there is a trend, it may have information about the primary source of sulfate from $SO_2$ emission or the rate of secondary sulfate production from $SO_2$ oxidation.

Page 11, lines 325-326: There is not a clear difference in Figure 4 between the blue / no alert trajectories and the non-haze trajectories. Are the authors sure that the 4 days are meaningful in this category to attribute the large contribution of industrial emissions?

In Figure 5, this category remains different from the trend in most other categories as the haze severity increases.

---

## Author Comment (AC1) · 27 Oct 2017

Correspondence to Review1

Thank you very much for your thorough and constructive comments on our manuscript acp-2017-515, entitled "Characteristics and source apportionment of fine haze aerosol in Beijing during the winter of 2013". We made all corrections and revised the manuscript according to your comments. The response is given to each comment. In the revised manuscript, changes are colored in blue.

General Comments

[Figure]

Comment 1: NMF is an approach which is less widely used by the community of PM source appointment than PMF or PMF/ME-2. It would be useful for the readers to judge the quality of the analyzed results if the authors could provide more details about the possible difference between NMF and PMF in the part of methods. It is well known that the use of such kind of statistical analysis tool is quite arbitrary. There are some plausible interpretations about the extracted factors in the paper. But please add uncertainty analysis of the NMF results.

Response 1: We agree with your view toward statistical analysis. PMF is more widely used than NMF for source apportionment for atmospheric particulate matter. Also, the result of statistical analysis is fairly arbitrary and should be interpreted with caution. In this study, we used NMF rather than PMF for the following reasons.

(1) Non-negative matrix factorization (NMF) is similar to positive matrix factorization (PMF) as mentioned in the text (Page 6 Line 158-161). Both methods find two matrices (W and H, termed the contribution matrix and the source profile matrix, respectively) that best reproduce the input data matrix (V) using the same factorization approach (V = WH) as a positive constraint (W$\geq$ 0 and H$\geq$ 0). However, difference between PMF and NMF lie in the method of treating negative factors and the algorithms which guarantee the solution matrices of W and H to be non-negative. When treating negative factors, PMF forces them to be positive, but in NMF only non-negative factors are used. It means that more tweaking is exerted to PMF, whereas less number of factors is extracted in NMF. If all conditions met, therefore, PMF analysis will provide more detailed information on sources, compared to NMF.

In addition, the additive update rule used in algorithms of PMF is applied to a multiplicative update rule for NMF method (shown below), which ensures the square root of the sum of squared differences of the elements to be non-increasing. Due to this improvement, the non-negative W and H matrices are initially guaranteed so that the tweaking of ad hoc non-negativities of PMF is not necessary at all for NMF (Lee and Seung, 2001). $W_i \leftarrow \{W_i (H_j^T V_{ij})/( H_j^T H_j W_i)\}$ $H_j \leftarrow \{H_j (W_i^T V_{ij})/(W_i^T W_i H_j)\}$

2) The uncertainty level is very important to PMF treatment. To calculate uncertainties, there are two methods employed for the EPA PMF 5.0 (User's manual, https://www.epa.gov/air-research/epa-positive-matrix-factorization-50-fundamentals-and-user-guide): observation- and equation-based uncertainty. The former requires an estimate of the uncertainty for each species in each sample. The observation-based uncertainty of components can be evaluated by repeated observations (cost a lot of time and resources) or by using several different instruments/methods (not available in this study) (see https://www.nist.gov/pml/nist-technical-note-1297). Hence, the equation-based uncertainty is usually used in PMF model, which provides species-specific parameters for each sample. The equation-based uncertainty can be calculated as follows:

$5/6 \times$ MDL (method detection limit) (concentration$\leq$ MDL). . . . . . .. . . . . . . . . . .1) [(Error Fraction $\times$ concentration)2 + (MDL)2]0.5 (concentration$>$ MDL). . . . . . . . . . . .2) , where error fraction (EF) is the percentage of uncertainty.

In Equation 2), uncertainty includes three terms, EF, concentration, and MDL, which is suitable for higher concentrations whereas Equation 1) is better for lower concentrations.

This study analyzed samples for winter season (three months), during which concentrations ranged from the level of detection limit for clean continental background to the extremely high level of severe haze event. For instance, $SO_4^{2-}$ concentrations varied from the detection limits to 100 $\mu$g/m3.

For PMF uncertainty calculation (e.g., Reff et al., 2007), the analytical uncertainty is the most critical factor. As stated in the text, carbonaceous concentrations were not directly measured but indirectly estimated in this study and thus, their analytical uncertainty is not available.

For source apportionment of PM2.5, therefore, we used NMF method with "0.3+DL" for estimating uncertainty according to the method of Xie et al. (1999a; b). In this

formula, a constant 0.3 corresponds to the log(Geometric Standard Deviation, GSD) to represent the variation of measurements. In the present study, concentrations of each species were converted into those of standard normal distribution. Then, log(GSD) was calculated from the normalized concentrations for all measured species, which was no greater than 0.3. Therefore, we adopted 0.3 for the uncertainty estimation. The unit of all measurements was set to $\mu$g/m3. This method has several advantages. First of all, one set of analytical/method detection limit with an additional additive term enables to avoid zero, which causes instability of factorization analysis (Xie et al., 1999b). In addition, the use of geometric standard deviation is suitable for our measurement set in a wide range of concentrations.

Using the NMF model, the five source profiles were extracted, with which we were able to distinguish major emission sources for the winter PM2.5 and haze aerosols of Beijing, even though the specific type of industry or secondary factors were not separated. Particularly, the sources apportioned by NMF analysis are well incorporated into the history of air masses estimated by backward trajectory analysis under gradual change in meteorological conditions (Fig. 5).

Xie, Y. L., Hopke, P. K., Paatero, P., Barrie, L. A., and Li, S. M.: Identification of Source Nature and Seasonal Variations of Arctic Aerosol by positive matrix factorization, J. Atmos. Sci., 56, 249–260, 1999a. Xie, Y. L., Hopke, P. K., Paatero, P., Barrie, L. A., and Li, S. M.: Identification of source nature and seasonal variations of Arctic aerosol by the multilinear engine, Atmos. Environ., 33, 2549-2562, doi.org/10.1016/S1352-2310(98)00196-4, 1999b.

Comment 2: The authors concluded that "To abate the severe haze in Beijing, therefore, it is necessary to reduce vehicle emissions in Beijing and further sulfur emissions from industrial complexes in surrounding cities." But this is not fully supported by the data presented in this work. Can you prove that local emissions are dominated by vehicles? Can you prove that sulfur emissions are mainly from industrial complexes in surrounding cities? How about the uncontrolled coal burning for sulfur emissions?

[Figure]

Response 2: Our conclusion is based on the measurements of SO2 and NO2 in conjunction with sulfate and nitrate, and comparison of their relative enhancement in several haze events under different meteorological conditions. This information is summarized in the Table below and given as supplementary information. From non-haze to red-alert haze, the portion of SO42- and NO3- against mass and the SO2/NO2 ratio increased, whereas fractions of mineral or salt species and trace elements decreased. Between non-haze and haze events, the increase of SO2 (18.7 to 36.9 ppb) was greater than that that of NO2 (26.8 to 50.2 ppb). During the three types of haze events, SO42- enhancement (4 to 32 %) was also greater than that of NO3- (16 to 31 %). These results demonstrate that the variation in concentration and fraction was greater for nitrogen than sulfur compounds depending on meteorological condition, which suggests the larger contribution of local sources to nitrogen than to sulfur.   Regarding uncontrolled coal burning, a recent study by Cheng et al. (2017) emphasizes its contribution to sulfur emission in Beijing region. The southern and eastern region of Beijing (Tianjin and Tangshan as stated in Page 11 line 314) were recognized as main source regions, from which haze forming air masses were transported to Beijing during orange- and red- alert haze in this study.

Spatial distribution of (a) PM2.5 and (b) SO2 emissions from household coal combustion in the BTH region in heating season of 2013 (Cheng et al., 2017).

In the wintertime of Beijing, air mass was usually transported from the northwest with high wind speed. What we observed in the present study is that as the high pressure system developed, winds were shifted westward and then gradually to the southwest and southeast. As a result, the stagnated condition was intensified and the haze-alert level was raised (Fig. 4). When air masses were rapidly transported from the northern area, no pollution alert was issued. As the air mass slightly lingered over the western regions, blue-alert haze occurred. With the air mass moved very slowly from the southwestern areas, orange-alert haze event lasted for three days. As the air was severely stagnated, the red-alert haze occurred in Beijing for five consecutive days,

when air was coming from the east. It is in accordance with the result of recent study, emphasizing the effect of meteorological condition on the severity of haze in Beijing (Cai et al., 2017) (added in revised manuscript of Page 11 line 316-318).

Cai, W., Li, K., Liao, H., Wang, H., and Wu, L.: Weather conditions conducive to Beijing severe haze more frequent under climate change, Nat. Clim. Change, 7, 257-262, doi:10.1038/nclimate3249, 2017.

Specific Comments:

1. In the part of introduction, the authors may add descriptions on the current alert system implemented in Beijing.

Response 1: More detailed information on the alert system of Beijing is given in IN-TRODUCTION with a relevant website for air pollution alert regulations (Page 3 line 63-67). The criteria are given in association with individual haze event in Page 7-8 line 197-207.

2. The term "pseudo-carbonaceous" in Figure 2 and other place of corresponding text sounds strange. Maybe the authors can use "Particulate organic matter".

Response 2: The "pseudo-carbonaceous" include EC as well as OC, even though OC concentrations are usually higher than those of EC. Because carbonaceous com-pounds were not measured, but estimated from other measurements in this study, it should be clarified. In this context, we employed the terminology "pseudo" in front of carbonaceous compounds.

3. Line 260 – 261 "This study was performed in winter, during which the chemical composition of PM2.5 was likely to be more dependent on source strength rather than photochemical oxidation," this argument is ambiguous. The secondary species like NO3- and SO42- must come from atmospheric oxidation processes. I think even in winter chemical composition of PM2.5 was also related to both source strength and oxidations. Also as shown in Figure 2, sulfate and nitrate were always dominating

chemical compositions especially for the conditions of pollution episodes.

Response 3: You are absolutely right that the oxidation reaction is important because its concentration was high during winter. Since SO2 and NOx emission are the greatest in winter and the least in summer, the source strength is the greatest in winter. The above statement is to explain the seasonal difference in the study region, comparing the amount of emissions and well–established photochemical reactions.

Indeed, the secondary formation encompasses various processes including photochemical oxidation in gas and aqueous phase and, homogeneous and heterogeneous reactions, which are still poorly understood.

In previous studies, Sulfur Oxidation Rate (SOR) [nSO42–/(nSO42–+nSO2)] and Nitrogen Oxidation Rate (NOR) [nNO3–/(nNO3–+nNO2)] used to be found high during summer (n represents molar concentration), which indicates the efficient conversion of SO2 and NOx to sulfate and nitrate, respectively. In this study, the average SOR and NOR were 0.14 and 0.12, respectively. While the average values were relatively low, these ratios were raised in haze events, particularly in red-alert haze (0.32 and 0.35, respectively), indicating enhanced contribution from secondary species.

In addition, high aerosol loading could impose reduction in radiation during winter haze event. Zheng et al., (2015) has reported that in Beijing, solar radiation dramatically decreased to 2.77 MJ m-2 d-1 during winter haze episode, compared to clean days (9.36 MJ m-2 d-1 on average). In addition, Wang et al. (2014) observed the background level of ozone concentration (< 10 ppb) in Beijing during winter heavy pollution days. The model showed a regional-scale reduction of ozone from 12~44 to less than 12 ppb and OH from 0.004~0.020 to less than 0.004 ppt. These results confirm that photochemical activity was weakened during haze events.

Recently, there has been increasing number of studies conducted in China, reporting the fast conversion of sulfate even in cold season and suggesting possible mechanisms for it (e.g., Wang et al., 2016). Liu et al. (2015) showed that homogeneous and

heterogeneous reactions were important to secondary production during haze days.

To avoid the confusion, therefore, this part in Page 9-10 line 260-267 and the relevant discussion was reworded with more detailed explanation as follows.

"This study was performed in Beijing during winter when primary emissions are the greatest. As Beijing is a megacity with its own emissions but also surrounded by big satellite cities with industrial complexes, it is apt to be affected by their emissions if meteorological conditions meet. In addition, the study period was characterized by frequent occurrence of severe haze, during which the major sources and the degree of aging were intimately coupled owing to distinct meteorological states. Therefore, these five factors primarily indicate direct emission sources with secondary production implicitly included."

Wang, Y., Yao, L., Wang, L., Liu, Z., Ji, D., Tang, G., Zhang, J., Sun, Y., Hu, B., and Xin, J.: Mechanism for the formation of the January 2013 heavy haze pollution episode over central and eastern China, Sci. China Earth Sci., 57, 14–25, 2014. Wang, G., Zhang, R., Gomez, M. E., Yang, L., Zamora, M. L., Hu, M., and Li, J.: Persistent sulfate formation from London Fog to Chinese haze, Proc. Natl. Acad. Sci., 113, 13630–13635, 2016. Zheng, G. J., Duan, F. K., Su, H., Ma, Y. L., Cheng, Y., Zheng, B., Zhang, Q., Huang, T., Kimoto, T., Chang, D., Pöschl, U., Cheng, Y. F., and He, K. B.: Exploring the severe winter haze in Beijing: the impact of synoptic weather, regional transport and heterogeneous reactions, Atmos. Chem. Phys., 15, 2969-2983, doi:10.5194/acp-15-2969-2015, 2015. Liu, X., Sun, K., Qu, Y., Hu, M., Sun, Y., Zhang, F., and Zhang, Y.: Secondary formation of sulfate and nitrate during a haze episode in megacity Beijing, China, Aerosol Air Qual. Res., 15, 2246-2257, 2015.

4. Line 262 – 264 "In addition, NO2 is more likely sourced from local emissions, but SO2 is expected to be transported from nearby regions." This is a good argument. But more discussions or evidences are required to support this argument.

Response 4: The response 3 is also relevant to this point. A table is given as supplementary information.

5. Line 273, what could be the high VOCs emitting industries? Please be more specific.

Response 5: The industrial processes using VOCs as raw materials such as furniture manufacturing, petroleum refining, machinery equipment manufacturing and printing (Wu et al., 2015). The description was added in Page 9 line 254-255 of revised manuscript.

Technical Comments: Line 202, 203 et al., I suggest the authors to present the concentrations of PM consistently for the significant figure as Line 177, 180 and 187, e.g. change 168.4 $\mu$g/m3 to 168 $\mu$g/m3.

Response: The significant figures were corrected in revised manuscript.

Please find figure in pdf file given as supplement.

Please also note the supplement to this comment:
https://www.atmos-chem-phys-discuss.net/acp-2017-515/acp-2017-515-AC1-
supplement.pdf

---

## Author Comment (AC2) · 27 Oct 2017

Correspondence to Review 2

Thank you very much for your thorough and constructive comments on our manuscript acp-2017-515, entitled "Characteristics and source apportionment of fine haze aerosol in Beijing during the winter of 2013". We made all corrections and revised the manuscript according to your comments. The response is given to each comment. In the revised manuscript, changes are colored in blue.

Specific comments

[Figure]

Comments 1: Page 3, line 53: Does this mean 3 – 16 days per year?

Response 1: Yes, it's for one year and "per year" is added in the manuscript (Page 3 line 53).

Comments 2: Page 3, line 56: Is there a quantitative estimate for the boundary layer depth?

Response 2: Yes, there is. In Zheng et al. (2015), the boundary layer depth was found to be reduced less than 100 m in pollution periods study (Page 3 line 56).

Comments 3: Page 3, line 62: suggest phrasing: "winter haze episodes are 5 days in duration" Comments 4: Page 3, line 72: replace "Over the past seven years (2000-2006)" with "Over a seven year period (2000-2006)". Then on line 75, add ": : : by 85% over this period". Comments 5: Page 6, line 157: negative rather than negatice

Response 3-5: According to your suggestions, we rephrased and corrected them.

Comments 6: Page 6, line 164: The uncertainty description is not clear. What are the units on "0.3 + the analytical detection limit" ? Is this a relative error, or does it have concentration units ?

Response 6: In the present study, we used NMF method with "0.3+DL" for estimating uncertainty according to the method of Xie et al. (1999a; b). In this formula, a constant 0.3 corresponds to the log(Geometric Standard Deviation, GSD) to represent the variation of measurements. In the present study, concentrations of each species were converted into those of standard normal distribution. Then, log(GSD) was calculated from the normalized concentrations for all measured species, which was no greater than 0.3. Therefore, we adopted 0.3 for the uncertainty estimation. The unit of all measurements was set to $\mu$g/m3. This method has several advantages. First of all, one set of analytical/method detection limit with an additional additive term enables to avoid zero, which causes instability of factorization analysis (Xie et al., 1999b). In addition, the use of geometric standard deviation is suitable for our measurement set in a wide

range of concentrations.

Comments 7: Page 7, lines 178-179: What is meant by "secondary standard of GB 3095-2012" ?

Response 7: GB 3095-2012 is the revision of the GB 3095-1982, which prescribe the "National Ambient Air Quality Standard" of China. In GB 3095-2012, the standard for PM2.5 was added. The word "secondary standard" is removed in the revised manuscript.

Comments 8: Table 1: Should the number of days with PM2.5/PM10 > 0.5 and < 0.5 add up to the total number of days with comparison to PM10? In other words, 47 + 47 does not equal 67. The text implies that it should (e.g., that 70% of the events were developed type, which would be 47/67). Is the correct number for PM2.5/PM10 < 0.5 = 20 ?

Response 8: Yes, the number of samples for PM2.5/PM10 < 0.5 is corrected to be 20 in Table 1 of the manuscript. It was an error.

Comments 9: Figure 3: The factors are shown on a log scale to illustrate the contributions from all of the components of chemical composition. However, the log scale hides the large contributions of individual components to each, such as sulfate to coal combustion. Can the figure also be shown on a linear scale for comparison to illustrate which components make large contributions to each factor? A linear scale would increase the contrast.

Response 9: The source profile of PM2.5 is shown in linear-scale below. As you mentioned, the contrast among factors are maximized in linear scale. However, the contributions from low concentrations are hardly seen in this plot. The concentrations of major constituents of atmospheric aerosols vary in wide range. For source apportionment, however, trace elements such as metals play a key role. Thus, it is quite typical to analyze source profiles in log-scale. In the present study, sulfate concentration was

raised up to 100 ïA▪g/m3 with metal concentrations remaining low during haze period. Thus, the original plots in log-scale are left in the revised manuscript.

[Composition linear-scale profiles of the five factors identified in NMF analysis]

Comments 10: Page 9, lines 241-243: Traffic is attributed to a factor with high nitrate an ammonium, with the ammonia precursor attributed to the same emission source as NOx, presumably. Should there also be an agricultural factor for the ammonia emissions? Can the authors comment?

Response 10: The agricultural or biogenic source for ammonia emission was not distinguished in this study. It is mostly because this study was performed in the megacity of Beijing (the region in the 5th ring) during winter. In other study conducted at the same location (CRAES in Beijing) in the winter of 2013 (Wang et al., 2016), the agricultural influence on ammonia was reported to be negligible, based on the measurement of stable nitrogen isotope ($\delta$15N). They also encountered severe haze events during the experiment period, during which the contribution from agriculture and biogenic source was negligible and the main contribution was from coal combustion and vehicle emissions.

Wang, Y. L., Liu, X. Y., Song, W., Yang, W., Han, B., Dou, X. Y., Zhao, X. D., Song, Z. L., Liu, C. Q., and Bai, Z. P.: Isotopic partitioning of nitrogen in PM2.5 at Beijing and a background site of China, Atmos. Chem. Phys. Discuss., https://doi.org/10.5194/acp-2016-187, 2016.

Comments 11: Page 9-10, lines 256-264: The authors suggest that secondary production is a relatively unimportant consideration. However, it is well known that sulfur oxidation rates in winter are typically slow, while NOx oxidation rates to NO3- can remain rapid (e.g., Calvert et al., Nature 1985). Can the authors comment on the source of sulfate? Does this likely arise from secondary oxidation of SO2, or does it rather come from a primary emission of more oxidized sulfur that leads to sulfate? An easy metric here would be the ratio of sulfate to SO2 in molar units. A similar comparison

could be given for NO3- to NOx.

Response 11: You are absolutely right that the oxidation reaction is important because its concentration was high during winter. Since SO2 and NOx emission are the greatest in winter and the least in summer, the source strength is the greatest in winter. The above statement is to explain the seasonal difference in the study region, comparing the amount of emissions and well–established photochemical reactions.

Indeed, the secondary formation encompasses various processes including photochemical oxidation in gas and aqueous phase and, homogeneous and heterogeneous reactions, which are still poorly understood.

In previous studies, Sulfur Oxidation Rate (SOR) [nSO42–/(nSO42–+nSO2)] and Nitrogen Oxidation Rate (NOR) [nNO3–/(nNO3–+nNO2)] used to be found high during summer (n represents molar concentration), which indicates the efficient conversion of SO2 and NOx to sulfate and nitrate, respectively. In this study, the average SOR and NOR were 0.14 and 0.12, respectively. While the average values were relatively low, these ratios were raised in haze events, particularly in red-alert haze (0.32 and 0.35, respectively), indicating enhanced contribution from secondary species.

In addition, high aerosol loading could impose reduction in radiation during winter haze event. Zheng et al., (2015) has reported that in Beijing, solar radiation dramatically decreased to 2.77 MJ m-2 d-1 during winter haze episode, compared to clean days (9.36 MJ m-2 d-1 on average). In addition, Wang et al. (2014) observed the background level of ozone concentration (< 10 ppb) in Beijing during winter heavy pollution days. The model showed a regional-scale reduction of ozone from 12~44 to less than 12 ppb and OH from 0.004~0.020 to less than 0.004 ppt. These results confirm that photochemical activity was weakened during haze events.

Recently, there has been increasing number of studies conducted in China, reporting the fast conversion of sulfate even in cold season and suggesting possible mechanisms for it (e.g., Wang et al., 2016). Liu et al. (2015) showed that homogeneous and

heterogeneous reactions were important to secondary production during haze days.

To avoid the confusion, therefore, this part in Page 9-10 line 260-267 and the relevant discussion was reworded with more detailed explanation as follows.

"This study was performed in Beijing during winter when primary emissions are the greatest. As Beijing is a megacity with its own emissions but also surrounded by big satellite cities with industrial complexes, it is apt to be affected by their emissions if meteorological conditions meet. In addition, the study period was characterized by frequent occurrence of severe haze, during which the major sources and the degree of aging were intimately coupled owing to distinct meteorological states. Therefore, these five factors primarily indicate direct emission sources with secondary production implicitly included."

Wang, Y., Yao, L., Wang, L., Liu, Z., Ji, D., Tang, G., Zhang, J., Sun, Y., Hu, B., and Xin, J.: Mechanism for the formation of the January 2013 heavy haze pollution episode over central and eastern China, Sci. China Earth Sci., 57, 14–25, 2014.

Wang, G., Zhang, R., Gomez, M. E., Yang, L., Zamora, M. L., Hu, M., and Li, J.: Persistent sulfate formation from London Fog to Chinese haze, Proc. Natl. Acad. Sci., 113, 13630–13635, 2016.

Zheng, G. J., Duan, F. K., Su, H., Ma, Y. L., Cheng, Y., Zheng, B., Zhang, Q., Huang, T., Kimoto, T., Chang, D., Pöschl, U., Cheng, Y. F., and He, K. B.: Exploring the severe winter haze in Beijing: the impact of synoptic weather, regional transport and heterogeneous reactions, Atmos. Chem. Phys., 15, 2969-2983, doi:10.5194/acp-15-2969-2015, 2015.

Liu, X., Sun, K., Qu, Y., Hu, M., Sun, Y., Zhang, F., and Zhang, Y.: Secondary formation of sulfate and nitrate during a haze episode in megacity Beijing, China, Aerosol Air Qual. Res., 15, 2246-2257, 2015.

Comments 12: Page 10, line 271: A large carbonaceous component is shown for blue

/ no alert days. However, there are only 4 days and 4 samples in this category. Is it possible that the deviation of the carbonaceous aerosol from the trend of decreasing contribution as the haze level increases is simply a result of the small number of samples in the blue / no alert category, leading to a statistically anomalous result? Can the authors comment on this?

Response 12: Since the experiment was carried out for 3 months in winter, the number of sample are not large enough to draw statistically significant results for each haze event. The haze event is very sensitive to meteorological condition, which shows large variability from year to year. Therefore, the purpose of this study is to better characterize haze events and to understand their causes. In this context, the large contribution from carbonaceous component is clearly a characteristic of blue alert haze for the study period but should be cautious about generalizing the result.

For better understanding, however, we provide a table comparing the average and standard deviation of pseudo-carbonaceous concentration for the entire and no/blue alert haze period. While the deviations are comparable, the average concentrations are different by four times. Therefore, it is likely that there is little chance in our result to be severely biased by the small number of samples.

[Comparison of carbonaceous concentration between no/blue alert haze and entire period]

Comments 13: Page 10, line 283 – 287: Following from the comment above, how does the sulfate /SO2 ratio vary as the haze alert level increases? Does this ratio increase, decrease, or stay the same? If there is a trend, it may have information about the primary source of sulfate from SO2 emission or the rate of secondary sulfate production from SO2 oxidation.

Response 13: As stated in Response 11, we examined Sulfur Oxidation Ratio (SOR) and Nitrogen Oxidation Ratio (NOR) for each episode, which is summarized in the table below. They are increased as haze alert-level increases. However, the SORs of the

haze events are I lower even in red-alert event, compared to those of warm season
(0.5~0.7) (Wen et al., 2016).

[The average SOR and NOR in different levels of haze alerts]

Wen, W., Cheng, S., Liu, L., Chen, X., Wang, X., Wang, G., and Li, S.: PM2.5 chemical
composition analysis in different functional subdivisions in Tangshan, China, Aerosol
Air Qual. Res., 16, 1651-1664, 2016.

Comments 14: Page 11, lines 325-326: There is not a clear difference in Figure 4
between the blue / no alert trajectories and the non-haze trajectories. Are the authors
sure that the 4 days are meaningful in this category to attribute the large contribution
of industrial emissions? In Figure 5, this category remains different from the trend in
most other categories as the haze severity increases.

Response 14: It is just 4 days for no/blue haze event but 57 days for non-haze days, of
which trajectories are pretty much scattered. Most of all, the duration of no/blue haze
is shorter than a day, for which one sample was taken for a day. Thus, it is highly likely
that all 4 trajectories for 24 hours don't correspond to haze occurrence. The difference
is better shown when averaging the 6-hour trajectories during the 4 no/blue haze days
and 57 no-haze days. These trajectories are compared in the figure below.

[Averaged backward trajectories of air masses for 3 days at 6-hour interval during
no/blue alert- and non- haze days]

Please fine tables and figures in the pdf file given as supplement.

Please also note the supplement to this comment:
https://www.atmos-chem-phys-discuss.net/acp-2017-515/acp-2017-515-AC2-
supplement.pdf